# Microplastics affect sedimentary microbial communities and nitrogen cycling

Meredith E. Seeley [1][✉], Bongkeun Song[1], Renia Passie[1] & Robert C. Hale[1]

Microplastics are ubiquitous in estuarine, coastal, and deep sea sediments. The impacts of microplastics on sedimentary microbial ecosystems and biogeochemical carbon and nitrogen cycles, however, have not been well reported. To evaluate if microplastics influence the composition and function of sedimentary microbial communities, we conducted a microcosm experiment using salt marsh sediment amended with polyethylene (PE), polyvinyl chloride (PVC), polyurethane foam (PUF) or polylactic acid (PLA) microplastics. We report that the presence of microplastics alters sediment microbial community composition and nitrogen cycling processes. Compared to control sediments without microplastic, PUF- and PLA-amended sediments promote nitrification and denitrification, while PVC amendment inhibits both processes. These results indicate that nitrogen cycling processes in sediments can be significantly affected by different microplastics, which may serve as organic carbon substrates for microbial communities. Considering this evidence and increasing microplastic pollution, the impact of plastics on global ecosystems and biogeochemical cycling merits critical investigation.

---

[1] Virginia Institute of Marine Science, William & Mary, P.O. Box 1346, Gloucester Point, VA 23062, US. [✉]email: meseeley@vims.edu

The increasing amount of plastic debris in the marine environment is a global concern. Consequences of large debris on individual organisms can be obvious (e.g., entanglement or intestinal blockage of a sea turtle or whale)[1]. Although microplastics (<5 mm) and nanoplastics (<1 μm)[2] are the most abundant forms of debris, elucidating their biological consequences is challenging and ecosystem-level impacts have not been well demonstrated[3]. Such ecosystem-scale effects could affect biogeochemical cycles and categorize microplastics as a planetary boundary threat[3,4]. Further, most studies have focused on microplastics in surface waters. Although many plastics are buoyant, microplastics are still exported to sediments after biofouling or incorporation into marine snow or fecal pellets[5,6]. In fact, an increasing number of studies have identified microplastics in freshwater, coastal and even deep sea sediments[7]. Hence, impacts on sedimentary communities and associated processes merit investigation. Although the interaction between floating plastic debris and microbes (forming a biofilm) has been well documented[8–10], to our knowledge only three studies to date have addressed biofilm formation on plastic in sediments. Nauendorf et al.[11] studied the surface colonization of plastic bags (polyethylene (PE) or a biodegradable polyester/corn starch composite) in organic-rich marine sediments. They observed rapid bacterial colonization of the polymer surfaces, but did not characterize the community composition. Harrison et al.[12] investigated the surface colonization of low-density PE (5 × 5 × 1 mm) in estuarine sediments. Using 16S rRNA gene clone libraries, they found that bacterial composition differed between sediment types (fine sand, medium sand, and silt) and that two genera (*Arcobacter* and *Colwellia*) comprised 84–93% of the total sequences identified[12]. The biofilm formation on large microplastics (3−4 mm in diameter) at the sediment water interface was explored by Pinnell and Turner[13] using shotgun metagenomics. These authors found that compared with PE terephthalate (PET), a bioplastic (polyhydroxyalkanoate, PHA) promoted growth of sulfate-reducing bacteria, whereas the biofilm of PET was not significantly different from a ceramic pellet control. These studies illustrate the ability of biofilms to form on plastic surfaces in sediments. However, an unanswered question is whether the addition of microplastics (a presumably recalcitrant carbon (C) pool) alters overall microbial community composition and biogeochemical cycling processes in sediments.

The impact of microplastics on sediment microbial communities may be particularly important in coastal salt marshes. These systems receive direct influx of microplastics from land runoff[14], poor waste management[15], storm drains and sewage overflows[16], and wastewater treatment plant outfalls[17,18]. Marsh vegetation and water circulation patterns promote the entrainment and deposition of suspended solids, organic matter (OM) and microplastics[19]. As such, coastal salt marshes are also extremely active zones of OM remineralization and biogeochemical cycling. Sediment microbial communities work in a depth-dependent cascade to remineralize OM. This digenesis typically starts with degradation of the most-labile OM within the thin, oxygenated layer and ends with fermentation of less labile OM in the anoxic zone. Of these microbially mediated, catabolic processes, denitrification is particularly important in removing excess reactive nitrogen (N) in coastal systems. Denitrification occurs in the suboxic zone and utilizes nitrate ($NO_3^-$) and nitrite ($NO_2^-$) instead of $O_2$ as the terminal electron acceptor in the oxidation of OM. Denitrification is nearly as energy efficient as the oxygen respiration pathway. It acts to remove N by converting $NO_3^-$ and $NO_2^-$ to gaseous N species, such as nitrous oxide ($N_2O$) and dinitrogen ($N_2$). An equally important reaction is nitrification, which occurs in the oxic layer, oxidizing ammonium ($NH_4^+$) to $NO_2^-$ and then $NO_3^-$. Denitrification activity is limited by $NO_3^-$ and $NO_2^-$ supply from in situ nitrification or anthropogenic sources. In general, these two pathways are critical for both the removal of excess N in polluted environments, as well as regulating productivity in N limited ecosystems[20]. The response of these inorganic N forms to microplastic pollution has only been addressed in two studies to our knowledge, neither of which evaluated the role of bacterial community composition in relation to nutrient fluxes[21,22].

In our study, we explored the effects of microplastics on the structure and function (specifically, nitrification, and denitrification) of microbial communities in coastal salt marsh sediments. Three common, petroleum-based plastics were chosen for testing: PE, polyvinyl chloride (PVC), and polyurethane foam (PUF). In addition, one biopolymer (polylactic acid, PLA) was included to compare the effects of a presumably biodegradable polymer with those more recalcitrant to degradation. To evaluate the potential for these plastics to influence sediment communities in the short-term, microplastics of these four polymers were added to individual sediment microcosms and incubated for 16 days (Fig. 1). Changes in the composition and diversity of sediment microbial communities were assessed based on MiSeq sequencing of 16S rRNA genes, whereas the functional genes in nitrification and denitrification were determined with quantitative polymerase chain reaction (qPCR). Dissolved inorganic N concentrations in the overlying water of sediments were measured to infer sedimentary N cycling processes. At the end of the microcosm incubation, a sediment slurry incubation experiment with $^{15}NO_3^-$ tracer was conducted to measure potential denitrification rates. From these results, we demonstrate that sediment microbial communities differentially respond to the addition of microplastics, with significantly different structural and functional responses occurring between polymer types.

## Results

**Microbial community structure.** A total of 1,379,639 sequences were obtained after merging and filtering raw data of 16S reads, with an average of 44,504 sequences per sample. Bacterial 16S sequences were predominant in each sample, with <2.21% archaeal 16S sequences, which were excluded in further analyses.

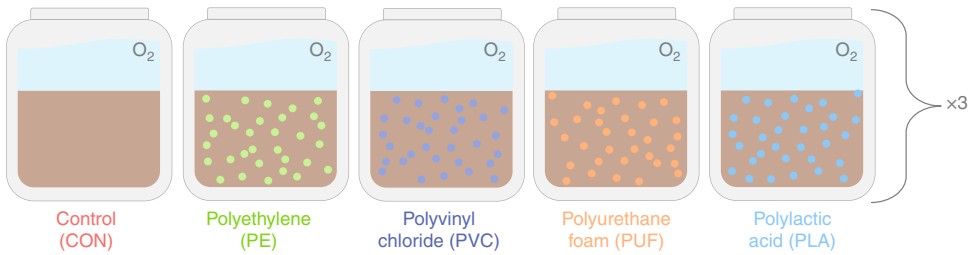

**Fig. 1 Microcosm experimental design.** All five treatments were repeated in triplicate and each microcosm was individually aerated to establish an oxygen gradient in the sediment.

**Table 1 Sediment community alpha diversity.**

|  | Shannon | Chao | Ace |
|---|---|---|---|
| **Initial** | | | |
| 0 days | 5.68 | 634.92 | 625.72 |
| **CON** | | | |
| 7 days | 5.8 ± 0.01 | 666.48 ± 9.92 | 660.16 ± 10.71 |
| 16 days | 5.79 ± 0.05 | 658.14 ± 32.09 | 655.48 ± ± 32.67 |
| **PE** | | | |
| 7 days | 5.55 ± 0.22 | 511.43 ± 174.91 | 510.54 ± 173.89 |
| 16 days | 5.75 ± 0.13 | 605.93 ± 103.29 | 603.21 ± 101.5 |
| **PVC** | | | |
| 7 days | 5.73 ± 0.06 | 724.42 ± 63.10 | 721.73 ± 63.22 |
| 16 days | 5.75 ± 0.02 | 657.45 ± 32.42 | 654.10 ± 32.08 |
| **PUF** | | | |
| 7 days | 5.89 ± 0.05 | 705.15 ± 60.32 | 701.10 ± 58.78 |
| 16 days | 5.82 ± 0.17 | 658.34 ± 130.66 | 656.86 ± 130.62 |
| **PLA** | | | |
| 7 days | 5.85 ± 0.02 | 668.62 ± 7.17 | 667.15 ± 6.45 |
| 16 days | 5.98 ± 0.03 | 790.98 ± 15.97 | 788.53 ± 16.92 |

Three diversity indices (Shannon, Chao and ACE) for bacterial communities within each sample day (0, 7 or 16) and treatment (n = 3), except the initial where n = 1. Values are included plus or minus standard error.

Bacterial diversity (alpha diversity measures) was highest in the biopolymer (PLA), lowest in PE-amended sediments and second lowest in the control treatment (Table 1; Supplementary Fig. 1). Bacterial community diversity was compared among samples using a principal coordinate analysis (PCoA), which measures dissimilarity among communities based on beta diversity (Fig. 2). Using Bray–Curtis dissimilarity, the first two principal components explained 32.7% of community variance. Multivariate permutational ANOVA (PERMANOVA) was used to calculate significant differences between these community dissimilarities based upon plastic treatment ($p = 0.001$), day ($p = 0.001$) and the interaction between these ($p = 0.023$; Supplementary Table 1). Sediment communities in the PVC treatment were distinctly different from the others. The initial community (sampled from the homogenized sediment upon experiment initiation) and the communities in control and PLA treatments clustered together in the top left quadrant. The communities in control and PLA exhibited minimal changes over time, according to the PCoA. PE and PUF treatments exhibited the most variation in community composition over time, but were similar to each other. These two petroleum-based polymer treatments had distinctly different effects on communities from the third petroleum-based polymer treatment, PVC. The clear differences in bacterial communities between PVC treatment and other treatments were also observed in a cluster dendrogram (Supplementary Fig. 2).

All samples were dominated by species within phyla *Bacteriodes* and *Proteobacteria* (Supplementary Fig. 3). Of the *Proteobacteria*, classes *Deltaproteobacteria* and *Gammaproteobacteria* dominated the communities (Supplementary Fig. 4). There were significant differences in community composition between treatments, particularly at the family level. The relative abundance of families at >1% in samples is illustrated in Fig. 3a (Supplementary Fig. 5 illustrates relative abundance of each sample). Significant differences in the relative abundance of these families between each treatment and the control (determined from DeSeq analysis; $\alpha < 0.01$; Supplementary Figs. 6–9) are illustrated in Fig. 3b. Several families showed a significantly higher relative abundance in the control than the PVC treatment community, including *Chromatiaceae*, *Ectothiorhodospiraceae*, *Lentimicrobiaceae*, *Magnetococcaceae*, *Pirellulaceae*, *Sedimenticolaceae*, *Thermoanaerobaculaceae*, and *Woeseiaceae*. Of these, *Chromatiaceae* and *Sedimenticolaceae* showed a significantly lower relative abundance in PVC-amended than all other treatments (Supplementary Fig. 4). *Family_XII* was significantly more abundant in communities of all plastic treatments than the control community. *Izimaplasmataceae*, *Marinifilaceae*, and *Marinilabiliaceae* exhibited a significantly higher relative abundance in the PE, PUF, and PVC treatments than the control, but not statistically more abundant than in the biopolymer (PLA) treatment. Several genera of *Desulfobacteraceae* and *Desulfobulbaceae* were significantly higher in the PVC-amended than the other treatments (Supplementary Figs. 9–12); this is not reflected in Fig. 3b because, although most genera within *Desulfobacteraceae* and *Desulfobulbaceae* showed a significantly higher relative abundance in PVC than the other treatments (Supplementary Fig. 9), at least one genus was also significantly lower, which resulted in the exclusion of those families from the heatmap. The most distinctly different treatment community, PVC, contained several families that showed a significantly higher relative abundance than all other treatment communities, including *Acholeplasmataceae*, *Anaerolineaceae*, *Family_XII*, *Izimaplasmataceae*, *Lachnospiraceae*, and *Marinilabiliaceae* (Supplementary Fig. 13).

**Nitrification and denitrification.** The concentrations of dissolved inorganic nitrogen forms (DIN), $NO_3^-$, $NO_2^-$, and $NH_4^+$, were measured in overlying water at each sampling point (Fig. 4). Concentrations of $NO_3^-$, $NO_2^-$, and $NH_4^+$ in the starting water were low (0.072, 0.527, and 3.44 μM, respectively). In general, concentrations of $NH_4^+$ were three times greater than $NO_3^-$ and $NO_2^-$ across all treatments, and there was two times as much $NO_2^-$ as $NO_3^-$. We observed greatest $NO_2^-$ and $NO_3^-$ in the 16-day PUF and PLA treatments, whereas $NH_4^+$ was lowest in these samples (Fig. 4). PE and control treatments had $NO_3^-$ and $NO_2^-$ in the water after 16 days, while PVC showed almost no detectable $NO_3^-$ or $NO_2^-$ at all time points. In contrast, $NH_4^+$ in the water was greatest in the PVC treatment after 16 days. In PE, PUF and PLA treatments, $NH_4^+$ was greater at 7 days than 16 days, opposite the PVC and control treatments. All statistical information can be found in Supplementary Tables 2–4. The $PO_4^{3-}$ water concentrations were also measured and were greatest in the PVC treatments (Supplementary Fig. 14, Supplementary Table 5).

The DIN concentrations can be used in conjunction with gene abundance to gain insights into nitrification and denitrification activities. Relative abundances of the genes involved in bacterial nitrification (*amoA*) and denitrification (*nirS* and *nirK*) were

measured based on qPCR of the targeted genes relative to 16S rRNA gene abundance. Ammonia monooxygenase (encoded by the *amoA* gene) is a critical enzyme in nitrification, oxidizing ammonia ($NH_3^+$) to hydroxylamine ($NH_2OH$). The ratio of

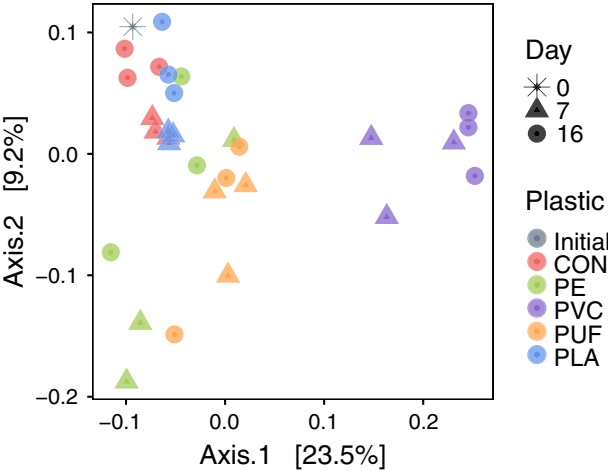

**Fig. 2 Principal coordinate analysis of the sediment communities.** Beta diversity was calculated using the Bray–Curtis dissimilarity index, and is plotted for all sample dates: initial (asterisk), 7 days (triangle), and 16 days (circle); and treatments: control (red; CON), PE (green), PVC (purple), PUF (orange), and PLA (blue). Significant effects of the plastic treatment ($p = 0.001$), day ($p = 0.001$) and interaction ($p = 0.023$) were tested by PERMANOVA.

*amoA* to 16S in different treatments is plotted in Fig. 5. We specifically targeted ammonia-oxidizing bacteria, not ammonia-oxidizing archaea (AOA), as no AOA taxa were detected in 16S sequences of the samples. Bacterial *amoA* gene abundances increased from day 7 to day 16 for all treatments, suggesting enhanced nitrification potential with time. The highest *amoA* gene abundances were in PLA and PUF treatments after 16 days (compared with control after 16 days, two-way ANOVA $p$-value $= 0.383$ and $0.0093$, respectively; see Supplementary Table 6 for all treatment comparisons), portending the highest nitrification activities. This was corroborated by the high $NO_3^-$ and $NO_2^-$ and low $NH_4^+$ concentrations in these samples, which are the products and reactants of nitrification, respectively. In contrast, *amoA* gene abundance was lowest in PVC treatment, which corresponds with the accumulation of $NH_4^+$ over time, indicating nitrification inhibition in this treatment.

A key enzyme in denitrification is nitrite reductase encoded by *nirS* and *nirK* genes. Denitrifiers carrying *nirS* genes are generally considered to be complete denitrifiers, converting $NO_3^-$ and $NO_2^-$ to dinitrogen ($N_2$); *nirK*-type denitrifiers are more likely to be incomplete denitrifiers, producing $N_2O$ as an end product and contributing to greenhouse gas emission[23]. The abundance of *nirS* and *nirK* genes (relative to bacterial 16S rRNA genes) showed very little variation over time within treatments (Fig. 5; Supplementary Tables 7–8). Control, PUF, and PLA treatments had the highest *nirS* abundances, suggesting a higher denitrification activity than the PE and PVC treatments. PVC consistently exhibited the lowest *nirS* gene abundances, suggesting a lower denitrification activity. Conversely, the *nirK* abundance was relatively consistent across all treatments, but slightly higher in the control after 16 days.

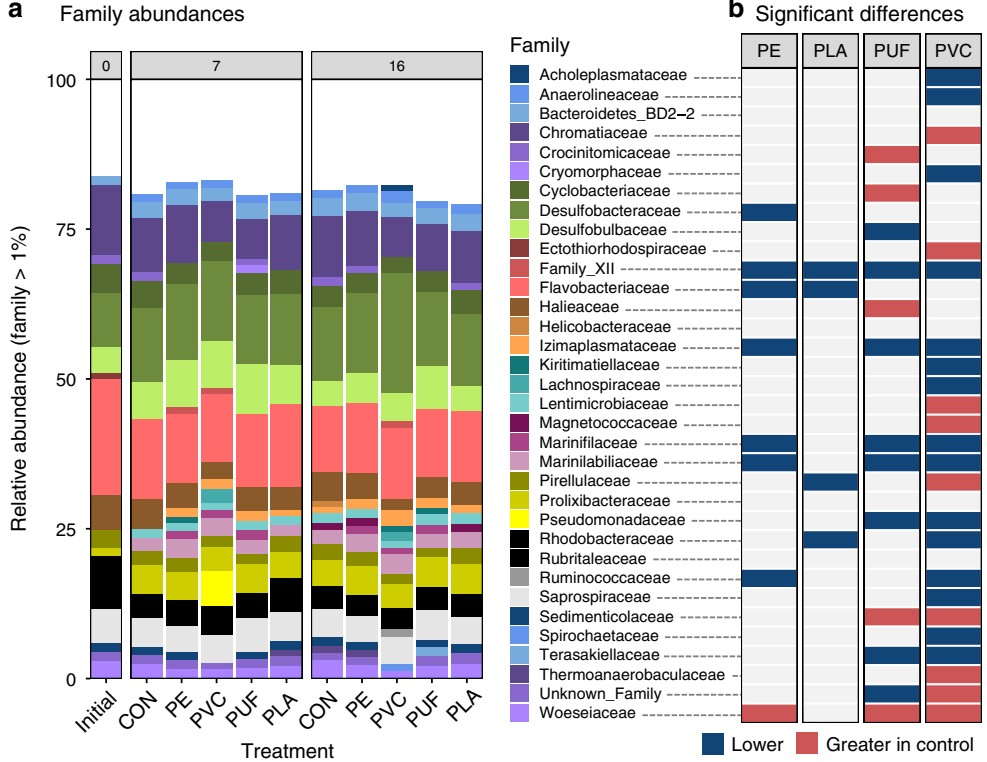

**Fig. 3 Bacterial community composition and treatment effects.** Comparison of taxonomic differences (family level) in bacterial communities with different microplastic treatments. Stacked bar plot of the relative abundance of families (>1% abundance) for each plastic treatment (averaged for the three replicates, $n = 3$ per treatment) for each sediment collection date (0, 7, and 16 days), where CON is the control treatment, **a.** Families that are significantly different in relative abundance between treatments and controls (averaged across collection dates), determined using DeSeq ($\alpha = 0.01$), is illustrated in **b,** showing if a family is significantly greater in one of the plastic treatments (blue) or the control (red).

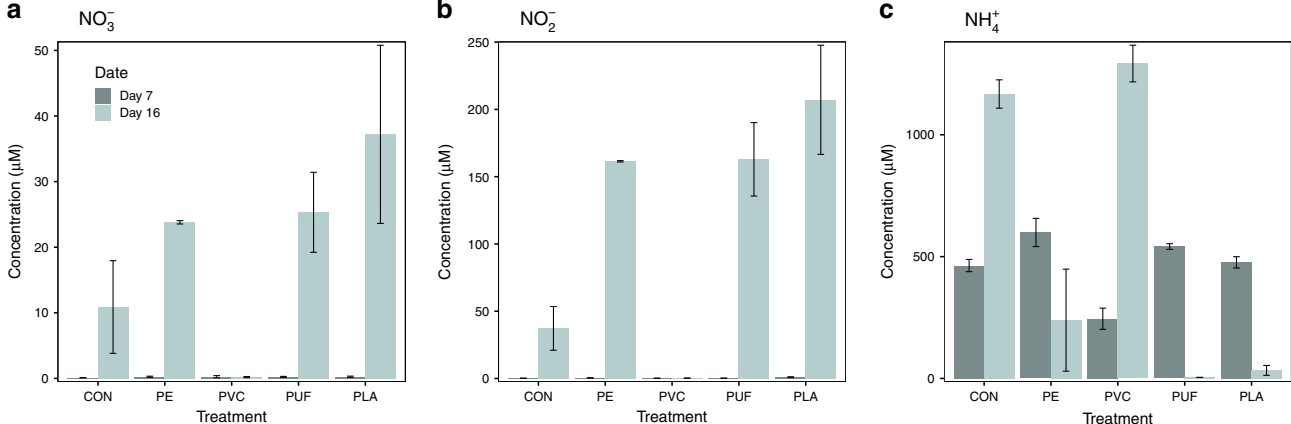

**Fig. 4 Dissolved inorganic nitrogen concentrations in water.** Concentrations (µM) of $NO_3^-$ **a**, $NO_2^-$ **b**, and $NH_4^+$ **c** are shown for each microplastic treatment and control microcosms after 7 and 16 days of incubation ($n = 3$ per treatment). Error bars are standard error and CON is the control treatment. Initial community ($n = 1$) concentrations are 0.072, 0.527, and 3.44 µM for $NO_3^-$, $NO_2^-$, and $NH_4^+$, respectively. Statistical analyses can be found in Supplementary Tables 2–4.

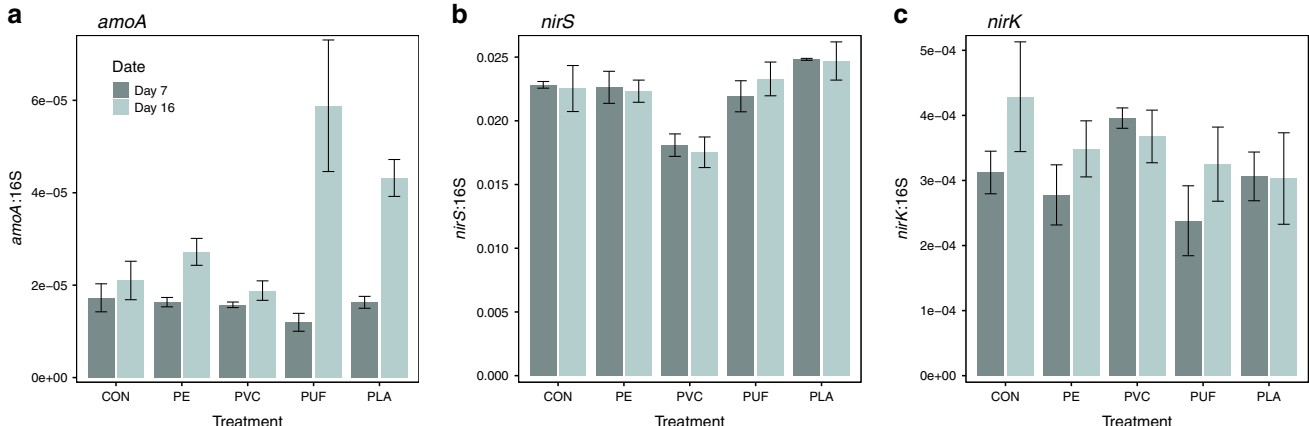

**Fig. 5 Nitrification and denitrification gene abundances.** The genes encoding ammonia monooxygenase (*amoA*, **a**) and nitrite reductase (*nirS*, **b**, and *nirK*, **c**) were quantified and normalized to 16S rRNA genes. Error bars are standard error ($n = 3$ per treatment) and CON is the control treatment. Initial community ratios are 1.85e−5, 3.03e−2 and 3.25e−4 for *amoA*, *nirS*, and *nirK*. Statistical analyses can be found in Supplementary Tables 6–8.

The potential denitrification activity rate was measured at the end of the 16-day incubation. PVC had a lower potential denitrification rate (Fig. 6; Supplementary Table 9) than any of the other microplastic treatments, coincident with the lowest *nirS* gene abundances. Denitrification was potentially highest in PLA and PUF, tracking their higher *nirS* gene abundances. PUF and PLA treatments also had more substrate for denitrification ($NO_3^-$ and $NO_2^-$). Interestingly, the control treatment had a significantly lower denitrification rate than PUF and PLA treatments, comparable to the PVC treatment. This deviates from the pattern of higher *nirS* and *nirK* gene abundances in the control than PVC treatment, but may be a product of relatively lower available $NO_3^-$ and $NO_2^-$ substrate compared with PUF and PLA. Potential rates of anaerobic ammonium oxidation (anammox) were also calculated in a subset of samples. Potential rates were highest in PLA and PUF, and lowest in PVC and the control (Supplementary Fig. 15; Supplementary Table 10), similar to denitrification. Sediment organic C and N contents were calculated at the end of the incubation (Supplementary Fig. 16), as well as the C and N of the plastics themselves (Supplementary Fig. 17), revealing that the control treatment was significantly lower in sediment organic C than all other treatments (Supplementary Tables 11 and 12). Further, the potential rates of denitrification and anammox were compared with total DIN, following Semedo and Song[24], to estimate DIN removal capacity. This revealed that PVC and the control treatments had the lowest DIN removal capacity, whereas PLA and PUF had the highest (Supplementary Fig. 18)[24].

## Discussion

Our study demonstrates that microplastic contamination affects both composition and function of sediment microbial communities. We report changes in the sediment communities between the control and plastic treatments, as well as differences owing to polymer type. These sediment communities encompass both the sediment in proximity to the microplastics, as well as the biofilms thereon. This unit is relevant as it reflects the overall impact of microplastic-contaminated sediments on the aquatic ecosystem. Further, attempts to physically separate the biological constituents of the microplastics and the sediment would disrupt these communities. The functional implications of these total sediment/biofilm changes were evaluated by monitoring DIN concentrations in the overlying water, as well as measuring relative abundances of nitrification and denitrification genes and post-incubation estimates of denitrification rate.

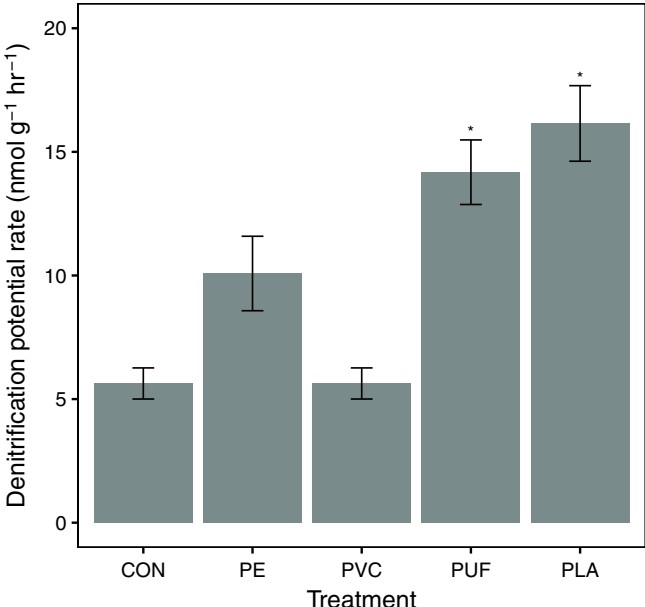

**Fig. 6 Comparison of potential denitrification rates.** Potential denitrification rate for each treatment in nmol g$^{-1}$ hr$^{-1}$, calculated after the end of the experiment (day 17). Error bars are the standard error ($n = 6$ per treatment) and asterix represent significant difference from the control ($p < 0.05$; Supplementary Table 9).

Based on both alpha and beta diversity measures, the different plastic treatments resulted in significant differences in the overall sediment bacterial community diversities (Table 1, Fig. 2). In all alpha diversity indices, the biopolymer (PLA) treatment was the most diverse and PE the least diverse community (Table 1). Although the PCoA explained <50% of the variance among treatments, there were clear deviations between PVC and the other petroleum-based polymer treatments. These are reflected in the significant differences between families present in those treatments (Fig. 3 and Supplementary Fig. 13), and motivated by the different polymer amendments. In contrast, the PLA treatment and the control were very similar. Although PE, PVC, and PUF polymers are all synthesized from petroleum-derived hydrocarbons, their compositions, structures and physical properties (i.e., strength, density, crystallinity, etc.) vary[25]. PE and PVC have C-C backbones, whereas PUF has a heteroatom in its main chain; further, PVC contains chlorine, whereas PUF contains N[26]. In addition, polymers may be amended with chemical additives to modify their properties to meet market demands[27]. Additive packages may be complex and often their compositions are withheld as confidential business information by the manufacturer. PE is the most abundant polymer in production and common in single-use containers[28]. In terms of marine debris, it is frequently reported in surface waters and increasingly in sediments, likely after its inherent buoyancy is overcome by biofouling[29]. PVC, on the other hand, is a high-density polymer, commonly used in industrial applications and construction. Phthalates may be present in products that require flexibility[27]. PUF is used in furniture, carpet underlayment, and insulation, and therein often contains percent levels of flame retardant additives[27,30]. Comparison of these common fossil fuel-based polymers to PLA, a heteroatomic biopolymer, is also warranted[31]. Biopolymers have been promoted as a more environmentally compatible alternative and may become a greater proportion of the market, and thus, of marine debris in the future. Indeed, Nauendorf et al.[11] reported that even biopolymers exhibited little degradation in organic-rich marine sediments. In summary, all four treatment plastics evaluated here vary in physical and chemical characteristics, hence their selection for this study. These differences contributed to the contrasting responses exhibited by the exposed bacterial communities.

We observed that microplastics generated from specific polymer types enhanced sedimentary nitrification and denitrification, whereas others inhibited these processes. In the case of PUF and PLA, in particular, there was an increase in $NO_3^-$ and $NO_2^-$ concentrations, a decrease in $NH_4^+$ concentration (Fig. 4), as well as correspondingly elevated *amoA* gene abundances with time (Fig. 5), suggesting enhanced rates of nitrification relative to the control. Presumably, the enhanced nitrification in these treatments depended upon $NH_4^+$ substrate. This may have been available through active sediment OM remineralization. In fact, $NH_4^+$ increased with time in the control and PVC treatments. This suggests remineralization was active, but nitrification might not be operating at a rate sufficient to remove this excess $NH_4^+$. Furthermore, in our microcosm experiment, nitrification and denitrification were coupled. Therefore, the increased $NO_3^-$ and $NO_2^-$ in PUF and PLA treatments may have facilitated the growth and activity of denitrifying communities, evidenced by higher *nirS* gene abundance (Fig. 5) and elevated potential denitrification rates (Fig. 6). For the same reasons, the PE treatments also appeared to slightly enhance nitrification and denitrification, although not significantly so. Furthermore, some forms of polyurethane have been reported to be susceptible to microbial degradation[32]. PUF contains nitrogen in the polymer backbone, unlike the other polymers tested here[26]. Theoretically, in situ degradation of PUF may have contributed to labile inorganic N for nitrification and coupled denitrification, and this possibility should be addressed in future studies.

In contrast, both nitrification and denitrification appeared to be inhibited in the PVC treatment. Not only were $NO_3^-$ and $NO_2^-$ concentrations in the overlying water extremely low, but the PVC treatment sediment also exhibited the lowest relative abundance of *nirS* gene and lowest potential rate of denitrification in the post-incubation sediment slurry experiment. Similar to the control, however, the PVC treatment had high concentrations of $NH_4^+$, which increased over time, likely owing to sediment remineralization. Thus, nitrification was clearly limited in this system. Sulfide has been documented to inhibit nitrification in marine sediments[33]. Most genera of *Desulfobacteraceae* and *Desulfobulbaceae* showed a significantly higher relative abundance in the PVC than the other treatments (Supplementary Figs. 9–12). Members of the *Deltaproteobacteria* class had highest relative abundance in the PVC treatment after 16 days, which is characteristic of sulfate reduction. Sulfide production in the PVC treatment by these abundant sulfate-reducing bacteria may have inhibited nitrification, and thus denitrification[33]. Pinnell and Turner[13] observed significantly higher sulfate-reducing microorganisms on the biofilm of a bioplastic (PHA) formed at the sediment water interface, compared to PET plastic and a ceramic control. They suggested that this was attributable to the hydrocarbon degradation of PHA by sulfate reducers. Here, however, sulfate-reducing bacteria were not observed in the biopolymer treatment (PLA), decreasing the connection between sulfate-reduction and hydrocarbon degradation. As such, the increase in sulfate-reducing bacteria in PVC remains unexplained. However, it is may be a function of the plastic composition (e.g., a shared additive between tested PVC and Pinnell and Turner's PHA) or a physical response of the sediment environment (e.g., increased hydrophobicity).

Insights into the mechanisms behind microplastic effects on the sediment microbiome and N cycling may be drawn from other studies. Cluzard et al.[22] observed an increase in overlying water $NH_4^+$ concentration when sediments were amended with

PE microbeads, similar to our PVC treatment. These authors proposed that an increase in sediment porosity allowed for greater diffusion from the sediments. However, increasing porosity would also increase oxygen diffusion and thus nitrification, decreasing $NH_4^+$. Indeed, our PVC treatment also exhibited high $PO_4^{3-}$ concentrations in the overlying water (Supplementary Fig. 14). This can be caused by decreased organic phosphorous burial and subsequent increased $PO_4^{3-}$ in the overlying water in some anoxic systems, which would be characteristic of a less-porous system[34]. Cluzard et al.[22] did not address community composition; thus, we cannot discern if sulfate-reducing bacteria, which could also have inhibited denitrification, were present in their samples. Another hypothesis is that the microplastics possessed antimicrobial properties, which may select for certain taxa (e.g., sulfate reducers and gram-negative[35]) and against others (e.g., nitrifiers[36])[37]. Plasticizer-containing PVC products with antimicrobial properties are often used in the medical field[38]. For example, Cluzard et al.[22] used pre-rinsed PE microbeads derived from a skin cleansing personal care product, which likely originally contained antimicrobial additives. In contrast, the PE used in our experiment was a pre-washed, composite of high- and low-density single-use, container-derived plastics. If microbial responses are indeed influenced by additive content and not polymer type alone, experimental and field research designs must consider both. We suggest future research to characterize additives, especially in controlled studies of organismal responses, so that their influence can be better assessed. If certain additives are found to inhibit coastal N cycling, their use in plastics could and should be controlled.

Compared with the plastic treatments, the control exhibited low denitrification activity following the incubation. This is contrary to denitrification genes (nirS and nirK), which were generally highest in the control treatment at 7 and 16 days. Over time, nitrification increased in the control (i.e., there was a slight increase in $NO_3^-$, $NO_2^-$, and amoA), thus denitrification was not limited by $NO_3^-$ and $NO_2^-$ substrate. Yet, partial denitrification contributing to the $NO_2^-$ pool could have occurred, in addition to nitrification, as evidenced by the considerably higher dissolved $NO_2^-$ than $NO_3^-$. The amount of organic C in sediments was notably different between control and plastic treatments, the latter receiving supplemental OM in the form of C from the polymer amendments (Supplementary Fig. 17). This suggests that the higher denitrification in plastic treatments, particularly PLA and PUF, may have been facilitated by the polymer OM itself. PLA and PUF are also the only heteroatomic polymers tested and more susceptible to hydrolytic cleavage, compared with the other plastics with solely C backbones[26]. However, again, other researchers have suggested that biodegradation of plastics in sediments is low[11]. Longer duration experiments should be performed to observe if bacteria can degrade plastic over time when faced with a labile sediment C limitation. Another factor for future consideration is preweathering of plastics, especially by photo-oxidation. This may prime them to subsequent biodegradation[26].

Clearly, plastic amendments also affected C cycling in our sediment microcosms. In aquatic environments, the bulk of plastic degradation studies have been on water column-originating biofilms of plastics and have focused on the presence of known hydrocarbon degrading species[9] or metabolic pathways[8]. In either case, the true capacity for bacterial plastic degradation and the responsible organisms are yet undetermined. Certain plastic-degrading species have been suggested for PE[10,39], PUF[10,39], PVC[39], and PLA[39]. Yet very few, if any, of these were found in our samples at >0.1% abundance (Supplementary Table 13). This is not surprising, however, as the above-cited studies are based on water column biofilm assemblages. In a study

addressing sediment microbial degradation of PUF, Shah et al.[40] reported the significant presence of Pseudomonas spp. We also observed these taxa in our PUF and PVC samples (Supplementary Table 14), including Pseudomonadales pseudomonadaceae, a previously reported petroleum hydrocarbon degrader in oil-polluted salt marshes[41]. Insight into novel, sediment-based, hydrocarbonclastic taxa may be inferred from operational taxonomic units (OTUs) that are significantly higher in plastic-amended treatments than the non-amended control. Family_XII Fusibacter was significantly higher in all treatments than in the control (Fig. 3). Families Marinifilaceae and Marinilabiliaceae were higher in all petroleum-based plastic treatments than the control and PLA treatments. No publications to our knowledge, however, suggest hydrocarbon degradation capacity of these organisms. Therefore, further research is needed. Nonetheless, the results from potential denitrification activity measurements suggest that plastics may be acting as an organic C source for sediment microbial communities (Supplementary Fig. 16). Our microcosm design provided no additional source of C substrate, which may have motivated sediment microorganisms to utilize microplastics as C for energy compared to natural systems.

Massive amounts of plastic enter and reside within riverine, estuarine and coastal environments. Although it was once considered completely recalcitrant, we now know that plastic degrades to varying extents in the marine environment over time and that microbial communities may play a role in this[9,25,42]. The leaching of chemicals from plastic alone has been shown to potentially contribute to the dissolved organic C pool in marine waters[43] and to the production of greenhouse gases, such as methane and ethylene[44]. It was estimated that between 1.15 and 2.41 million tons of plastic enter the coastal zone and oceans from rivers annually, much of which eventually reaches sediments[45]. These plastics once served a variety of consumer purposes; as such, they are extremely diverse in form and chemistry. Here, we have demonstrated that microplastics generated from four diverse polymers influenced marsh sediment microbiomes and biogeochemical cycling. Although the difference between biofilm communities and that of the surrounding sediment cannot be differentiated using our approach, the outcomes between our treatments robustly illustrate the influence microplastics may have on intact sediment ecosystems. This is foundational for future efforts to assess risks of microplastic pollution in diverse environments. Further, the work presented here demonstrates that microplastics are indeed capable of ecosystem-level effects, including alteration of biogeochemical cycles[3]. Thus, we should evaluate plastic debris as a potential planetary boundary threat[3,4,46].

## Methods

**Experimental plastics**. Consumer plastics were milled and sieved to a defined size range, 53–300 μm. PE was a recycled product of predominantly high-density PE obtained from Envision Plastics (Reidsville, NC). PVC used consisted of yellow pellets from Teknor Apex. The PUF was a flexible, yellow PUF donated from a gymnastics studio, similar to PUF used in furniture cushioning. The PLA pellets were from IC 3D Printers LLC and are commonly used in 3D printing. All plastics were embrittled and ground to a powder using a Retsch CryoMill. Resulting powders were individually sieved to 53–300 μm using a Retsch AS 200 air jet sieve. In previous studies, PUF, PVC, and PE were analyzed for flame retardants. PVC was tested for phthalate additive content (see Supplementary Methods and Supplementary Table 10). Previous analysis of the PVC used here revealed diethylhexyl phthalate at 8.61 mg g$^{-1}$. PUF used in this study contained both brominated and phosphate-based flame retardants (Supplementary Table 15). This additive analysis is not comprehensive, however, and does not include PLA. Further, being a foundational study in the field, the plastics we selected were not intended to be representative of all environmental sediment microplastic pollution, but rather to embody a range of the characteristics that may be encountered.

**Sediment microcosm incubation**. Sediment was collected at low tide from the top 2 cm of an intertidal marsh, located along the York River estuary in Gloucester

Point, VA in March 2018. Sediment was sandy, with low organic C content (Supplementary Fig. 16). It was thoroughly homogenized and interstitial water and large debris removed. An aliquot was sampled and immediately frozen for initial community analysis ($T_0$). Approximately 300 g of wet sediment were added to acid washed and combusted 500 ml glass jars (sediment depth reached 3 cm). The experimental design included four microplastic (53–300 μm) treatments (PE, PUF, PVC, PLA) and a no plastic control (CON), with three replicates each ($n = 15$; Fig. 1). Microplastics were added to obtain a concentration of 0.5% by weight of sediment, or 1.5 g of microplastics per microcosm (300 g sediment), and thoroughly homogenized with the sediment prior to adding water. Published field data on microplastic sediment concentrations are limited. Many studies that report microplastics in sediments do so on a particle count (not weight) basis, making comparison with our microcosms impractical. Reports generally underestimate actual burdens as they do not include small microplastics. Carson et al.[47] reported the weight-based sediment concentration of microplastics to sediment for a Hawaiian beach to be 3.3%, six times higher than our experimental concentration Another Hawaiian study reported concentrations closer to 0.12% plastic by weight (reported as 2 g L$^{-1}$ and converted using 1.7 g cm$^{-3}$ for marine sediment density[48]), five times lower than our experimental concentration[49]. Based on these available studies, we believe our exposure concentrations are relevant, particularly for foundational work.

Estuary water was collected adjacent to the sediment sampling location along the York River (salinity: 21) and filtered with a 38 μm pore size filter to remove particulate matter. Filtered water (50 mL) was added to each microcosm, mixed, and sediments allowed to settle. All samples were carefully topped with an additional 200 mL of filtered seawater, so not to re-disturb the sediment. Microcosms were gently aerated to maintain oxygen in the overlying water and 24 hours were allowed to establish an oxic/anoxic gradient down the sediment prior to the start of incubation period. Microcosms were covered with aluminum foil to prevent evaporation and maintained at room temperature in the dark during incubation.

Sediment aliquots were collected at 7 and 16 days for microbial community analysis. Triplicate cores (depth: 1 cm; diameter: 0.5 cm) were randomly sampled at three locations within the microcosm. The three cores were homogenized, centrifuged and supernatant water removed by pipet. Sediment composites for each microcosm and sample date ($n = 30$) were stored in a $-60$ °C freezer for DNA extraction. Coincident with sediment sampling, 10 mL of overlying water were collected and immediately frozen for inorganic nutrient analysis.

**DNA extraction and 16S rRNA gene analysis.** A DNeasy Powerlyzer Powersoil Kit (Qiagen) was used to extract DNA following the manufacture's instruction. In brief, silica bead tubes were loaded with 0.5 g of sediment and extraction solution, using the bead beater to break cells and extract DNA. Microplastics were not removed from the sediments prior to DNA extraction. The supernatant in each tube was purified and prepared with a series of DNA-cleaning solutions, and final DNA was eluted in 50 μL. Qbit fluorometric quantification (Thermo Scientific) was used to measure the extracted DNA, and each sample was diluted to 10 ng μL$^{-1}$. Diluted DNA (1 μL) was combined with 12.5 μL GoTaq mix, 9.5 μL nucleic acid free water and 1 μL each PCR primers (515 F and 926 R) to target the V4-5 regions of 16S rRNA genes[50]. PCR was carried out with denaturation at 95 °C for 3 minutes, 25 annealing cycles at 95 °C for 30 seconds, 55 °C for 30 seconds and 72 °C for 30 seconds, followed by elongation at 72 °C for 4 minutes. The PCR product was purified using an AMPure XP bead kit and the concentration calculated using Qbit fluorometric quantification. All PCR products were diluted to 0.2 ng μL$^{-1}$ and 6 pM of this product was used for sequencing with the Illumnia MiSeq platform, following the manufacturer's instruction. All genes were normalized to the 16S qPCR concentration, to correct for nucleic acid concentration.

The high-quality sequences from the Illumina MiSeq were processed using dada2 plugin for RStudio[51]. In brief, forward and reverse sequences were trimmed to 200 and 250 base pairs and a maximum error number of 2 and 5 errors, respectively. Sequences were merged and aligned, and chimeras removed. The Silva reference database (version 132) was used to match the taxonomy information of sequences[52]. Code is provided in the Supplementary Code. RStudio packages (phyloseq, ggplot2, and vegan) were used for all graphical and statistical analyses (McMurdie and Holmes, 2013).

**Quantitative PCR of targeted genes.** QPCR assays of 16S rRNA, amoA, nirS, and nirK genes were conducted using the QuantStudio 6 Flex (Thermo Scientific), as described by Lisa et al. and Semedo et al.[24,53]. Standards were prepared through a serial dilution of M13 PCR products from plasmids carrying the target gene or fusion PCR products from environmental DNA and quantified using an Agilent 220 TapeStation System (Agilent Technologies). The primers used for qPCR of 16S rRNA genes were 515 F and 926R[50]. The primers nirScd3aF and nirSR3cd were used to generate 400 bp amplicons of bacterial nirS genes; nirK genes were detected using nirKF1Ac and nirKR3Cu primers[54]; bacterial amoA genes were detected using AmoA-1F and AmoA-2R[53]. The 12 μL qPCR reactions for 16S, nirS, nirK, and amoA quantification consisted of 6 μL of SYBR green GoTaq qPCR Master Mix (Promega), 0.03 μL of CRX dye, 0.6 μL of each primer (10 μM), 0.12 μL of bovine serum albumin, 8 ng of template DNA, and were adjusted to final volume with nuclease-free H$_2$O. The qPCR conditions can be found in reference

publications for 16S[24], nirS, nirK, and amoA[53]. Amplification efficiencies were 69%, 76%, 74%, and 84%, for 16S rRNA, nirS, nirK, and AmoA genes, respectively. The $R^2$ value of the standard curves was 0.99 for the four genes. All reactions were performed in 384 well plates with three negative controls, which contained no template DNA, to exclude potential contamination. Reaction specificity was confirmed using gel electrophoresis in comparison with standards and monitored by analysis of dissociation curves during quantitative amplification. Gene ratio of amoA, nirS, and nirK genes in different treatments was calculated by dividing the gene copy numbers by bacterial 16S rRNA gene copy numbers.

**Rate measurements of denitrification and anammox.** Sediment slurry incubation experiments, with $^{15}NO_3^-$ as a tracer, were conducted after 17 days incubation time, with exetainer tubes for each treatment replicate ($n = 6$ per treatment) following Lisa et al.[53] In brief, exitainer tubes with 2 g of homogenized sediment were helium-purged and dark-incubated overnight to remove residual $NO_2^-$ and $NO_3^-$. Six replicates of exetainer tubes per sample were amended with 100 nmoles $^{15}NO_3^-$ and then incubated at room temperature in dark. Both anammox and denitrification activities were stopped by adding saturated zinc chloride (ZnCl) solution after 0, 1, and 2 hr of incubations. Time series production of $^{29}N_2$ and $^{30}N_2$ was measured on an isotope ratio mass spectrometer and used to calculate the rate of denitrification and anammox following Song and Tobias[55].

**Sediment and water column nutrients.** Water samples from each collection date (including the initial water, $n = 31$) were analyzed for $NO_2^-$, $NO_3^-$, $NH_4^+$, and $PO_4^{3-}$ content using a Lachat QuickChem 8000 automated ion analyzer, per methods in Anderson et al.[56]. In addition, total organic carbon and nitrogen content were analyzed by the Virginia Institute of Marine Science Analytical Service Center using an Exeter model 440CE CHN analyzer.

**Statistical analyses.** Differences in rate, gene abundance or nutrient concentration between treatments were statistically compared using a one-way or two-way ANOVA ($\alpha < 0.05$) in RStudio[57]. Prior to analysis, the Shapiro–Wilks test for normality and Levene's test for homogeneity of the variance were conducted. A post hoc Tukey test was used to determine which treatments were significantly different. A multivariate PERMANOVA was conducted using the anodis function (Vegan package, Rstudio) to evaluate significant effects of plastic, date, and the interaction of these on community dissimilarity. Results of all analyses may be found in Supplementary Tables 1–12.

**Reporting summary.** Further information on research design is available in the Nature Research Reporting Summary linked to this article.

## Data availability
The 16S sequence data that support the findings of this study are available in NCBI repository (accession code: PRJNA575886). The authors declare that the remainder of data that support the findings of this study are available within the article and source data file.

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

## Acknowledgements

We thank the Freeman Family Foundation for support, via the Virginia Institute of Marine Science (VIMS) Freeman Family Fellowship. This work was partially supported by National Science Foundation grants (grant numbers: 1737258 and 1658135) and the National Oceanic and Atmospheric Administration (grant number: NA13NOS4630062). We also appreciate a Plumeri Faculty Excellence Award from William & Mary and VIMS Academic Studies department for APC support. This is contribution #3890 from the Virginia Institute of Marine Science, William & Mary.

## Author contributions

M.E.S. and B.S. designed the experiment. R.C.H contributed all plastics and their associated analyses. M.E.S. completed 16S rRNA gene extraction and analysis. R.P. completed qPCR and, denitrification and anammox experiments. Nitrogen cycling and DNA analysis was completed under the supervision of B.S. and in his lab. M.E.S completed first draft and figures of this manuscript. M.E.S, B.S., and R.H. contributed to manuscript editing and interpretation of data and results.

## Competing interests

The authors declare no competing interests.
