## [Peer Review File · Nature Communications]

Reviewers' comments:

Reviewer #1 (Remarks to the Author):

The authors explore the environmentally important question of potential impacts of microplastic (MP) pollution within aquatic ecosystems and in this case, upon sediment microbial communities and the additional potential impact of MP upon key biogeochemical cycling processes, in this case nitrification and denitrification. The authors utilise microcosm experiments in concert with molecular analysis of microbial communities and functional guilds and investigation of changes in concentration of reactive nitrogen species. The research questions are of interest and the data presented highlights some potentially interesting findings relating to variation in community composition and/or nitrogen biogeochemistry between different polymers. Whilst these data are intriguing, the interpretation and associated statistical analysis of the data and related discussion does not always support the conclusions being drawn, in particular due to challenges in experimental and sampling design. The points below highlight the key areas of concern and/or where further information and analysis would be necessary to support the authors's claims.

1. L 33. Please see Pinnell & Turner (2019) *Frontiers Microbiol.* is very relevant to the current manuscript.
2. L55 excess reactive nitrogen
3. L59 Nitrification is not the counterpart to denitrification. Nitrification converts ammonium to nitrate, denitrification converts nitrate to nitrogen.
4. L86-87 The authors state they exclude the archaea, but later in the text (L146) they talk about archaeal sequences not containing any archaeal taxa considered to be ammonia oxidisers. This is conflicting. I'd also suggest that testing by PCR for presence of archaeal *amoA* genes would nevertheless be essential, to categorically exclude the presence/role of archaeal ammonia oxidisers.
5. L91-100 & Figure 2. Variation between communities in treatments needs testing statistically. Use of ANOSIM or similar to test for treatment and/or time-specific variation is needed. The PCoA does show clear separation of communities in the PVC treatment from other treatments, but the subsequent Figure S17 suggests a potential reason for this, namely the presence of biofilm on the surface of these sediment microcosms. Unfortunately, the experimental/sampling design has not allowed partitioning of community composition between sediment vs. microplastic (in sediment), and in the case of the PVC treatment, there would seem to be a third community present in these microcosm (the surface biofilm). Unfortunately, without physical separation of the microplastics at the end of the experiment from the sediment, it becomes impossible to determine whether differences in community composition between substrates are driven by differences in the communities in the sediment and/or on the biofilms present on each microplastic type. In the PVC treatment, there may be a third community (ie. the biofilm), and this may drive the separation of the PVC (overall) community to diverge from those in other treatments.
6. L107-124 and Figure 3 and associated Supplementary Figures. What is not clear from the presentation of this data is whether any of these taxa are major components of the communities. One of the problems is that as data is averaged across replicates, we can't see whether an individual replicate may be significantly skewing increases/decreases in a taxon within a treatment. Additionally, the increases presented could be relatively small (eg. an increase in taxon abundance from 1% to 2%), or much more important eg. a 10-fold increase of 1% to 10%. For this description of data, the order of magnitude and in particular if this relates in a taxon becoming a major component within a community is of interest and this description should drill down on those taxa which become important / abundant (or conversely) that may decline following treatment (ie. be negatively impacted by a particular polymer).
7. Figure 4. Need to recognise that formation of nitrite could be via both ammonia oxidation and/or partial denitrification.
8. L161 and associated text on Q-PCR. Need to determine whether differences in relative abundance of these genes is significant between treatments and/or timepoints.
9. L166 and Figure 6 and SIFigure16 (Rate measurements). Insufficient detail is given in the Methods to allow interpretation of the denitrification rate measurements. The authors also mention data for

measurement of anammox (although there are no corresponding methods). Far more information is needed on this and additionally for anammox ¹⁵N-labelling of both ammonium and nitrite is needed.

10. L184 I've highlighted this already in Point 5 above, but the sampling regime doesn't allow discrimination between the communities in the sediment vs. the communities that may form biofilms upon the substrates. Potentially, community shifts could be due to substrate-specific communities on the microplastics, rather than changes in the bulk sediment. As noted previously for the PVC treatment, the community shift may be driven by the sub-community present in the biofilm.

11. L224-229. This is problematic. Depending upon the sediment type/composition (which also needs characterisation and presenting), it is possible that denitrification may only be occurring in the top 1-5 mm. By homogenization of samples during processing, the authors have lost the ability to distinguish variation in community composition (and function), between different redox zones. (see also L237-239).

12. L229-231. This is highly speculative.

13. L243. Claim of increases in sulfate reduction being a product of plastic composition needs justification.

14. L247-248. If, as the authors suggest, presence of microplastics increases sediment porosity, this should lead to increased oxygenation which should stimulate (aerobic) nitrification and hence potential reduction of ammonium concentrations rather than increases.

15. L254-255. Why should antimicrobial properties of microbeads be active against nitrifiers specifically and not against other guilds and/or taxa?

16. L290-314. The long discussion on hydrocarbonoclastic taxa and potential hydrocarbon degradation is tangential to this current study and should be removed.

17. L343. I'm unclear as to whether there was any attempt to sterilise plastics going into the experiments. For example, for the PUF, from the description, I'm assuming this is used-polymer and potentially already colonised by microorganisms.

18. L355. Needs characterisation and description of sediment to better understand experimental system.

19. L401-402. Normalization to Q-PCR 16S rRNA gene numbers is not clear.

20. References. Some references are incomplete eg lack page numbers. Others have inconsistent format usage (eg. inconsistent use of capitals in article titles), use of DOI rather than volume/page numbers for long-published papers (eg. Ward, 2008), gene names need italicizing etc.). Please review/revise all references.

Reviewer #2 (Remarks to the Author):

Author provides an interesting experiment about adhered microorganisms on MPs and nitrifying and denitrifying activities under exposure experiment. Similar exposure experiment has been addressed in previous reference (Wei et al., 2019). Therefore, I don't think it is novel and sound enough to publish in Nature Communication

Below, I only outline some major concern.

Line 341-353. Why author used commercial plastics in present study and was it consistent with environmental plastic debris in terms of weathering, composition and size range?

I think it would be more convincing.

Line 354 incubation was performed in the lab with too many controlled factors. It is hard to believe to reveal the actual reaction.

Minor comment

Line 27-30. This sentence should be reorganized to clearly present the intended meaning

Reviewer #3 (Remarks to the Author):

The manuscript by Seeley et al analyzed the bacterial community structure as well as abundance and activity of N-cycling bacteria in a salt marsh sediment in response to additions of different plastic materials. The authors performed a microcosm experiment adding four different types of plastic to the incubations and determined the rates for potential denitrification and the anammox process.

Molecular analyses (16S sequencing, qPCR for bacterial *amoA*, *nirS*, *nirK*) have been used to investigate the relative abundance of prokaryotic taxa and to quantify corresponding microbial guilds involved in N-cycling.

The study is a valuable contribution to our understanding of the relative impact of different (micro) plastics on bacterial community structures and overall N-cycling in salt marsh ecosystems.

Comments:

- The experimental set up is not complicated, therefore I think Fig. 1 is not essentially needed. Move to supplement.
- In Table 1 it would be good to include the errors for the indices. That way its easier to understand differences between days and treatments.
- line 111: In the figure caption please correct to: "Families were determined to be significantly different between treatments..."
- Figures 4, 5, S13, S14, S15, S16: If I am not mistaken the minimal number of data points for a boxplot is 9. If you have less, you should rather show all individual data points as the visualisation as a mean with spreads is misleading. Please also make sure to always include a statement on the number of n in the figure legends. This is missing in a few cases. In case of S16, where n=1, I recommend a table instead.
- As an addition to Fig. 4 I suggest to calculate the N-balance with the data on inorganic N. This way you could see if any substantial amounts of N were lost from your system (e.g. through denitrification) or gained (e.g. trough N mineralization). Also, have the initial values been measured? If so, please include in this graph it may make trends/ dynamics more clear.
- Fig 5: I would prefer to see the data for the initial time point in the graph rather than wrapping my head around a double ratio, which is what the normalization to 16S and initial community is I presume. So please just show all 16S normalized data.
- Fig. 6: Please include the unit for the y-axis directly at the axis. Also, add letters to indicate sig. differences to the control.
- line 223-224: How do you arrive at this statement? The conditions for the potential denitrification assay were quite different than the conditions of the microcosm experiment.
- line 233: Please correct: "... in the overlying water extremely low, ..."
- line 238: Please rephrase to something like: Most genera of [...] showed a significantly higher relative abundance in... This should be carefully corrected in other places of the manuscript too.
- In the method section the description of the qPCR assay for the *amoA* gene is missing. Please amend.
- line 439-441: In the listed reference NO₂⁻ and NO₃⁻ are not measured separately, but in this manuscript there is data specifically for both. Furthermore, PO₄³⁻ is not mentioned in the reference at all. Please correct and mention method for determination of NO₂⁻, NO₃⁻ and PO₄³⁻ quantification.
- Fig. S1: Please add error bars and indicate from which day of the incubation the data was obtained.

Reviewer #1 (Remarks to the Author):

The authors explore the environmentally important question of potential impacts of microplastic (MP) pollution within aquatic ecosystems and in this case, upon sediment microbial communities and the additional potential impact of MP upon key biogeochemical cycling processes, in this case nitrification and denitrification. The authors utilise microcosm experiments in concert with molecular analysis of microbial communities and functional guilds and investigation of changes in concentration of reactive nitrogen species. The research questions are of interest and the data presented highlights some potentially interesting findings relating to variation in community composition and/or nitrogen biogeochemistry between different polymers. Whilst these data are intriguing, the interpretation and associated statistical analysis of the data and related discussion does not always support the conclusions being drawn, in particular due to challenges in experimental and sampling design. The points below highlight the key areas of concern and/or where further information and analysis would be necessary to support the authors's claims.

Thank you sincerely, for your time reviewing our article. We appreciate the insight you provided, and feel it has improved the quality of our paper. In response to some of your comments, we feel we may have poorly presented some information (particularly, the rationale behind our experimental design). We have explained this and amended the manuscript according to the other points you made. We address each point individually, below. Thank you again, for your time and consideration!

1. L 33. Please see Pinnell & Turner (2019) *Frontiers Microbiol.* is very relevant to the current manuscript.

Thank you for referencing this article for us, and we lament that we had not included it in our original submission. Information from this paper has been included in our current paper, specifically in lines 44-55 and 434-443.

2. L55 excess reactive nitrogen

This has been modified accordingly.

3. L59 Nitrification is not the counterpart to denitrification. Nitrification converts ammonium to nitrate, denitrification converts nitrate to nitrogen.

This has been modified accordingly.

4. L86-87 The authors state they exclude the archaea, but later in the text (L146) they talk about archaeal sequences not containing any archaeal taxa considered to be ammonia oxidisers. This is conflicting. I'd also suggest that testing by PCR for presence of archaeal amoA genes would nevertheless be essential, to categorically exclude the presence/role of archaeal ammonia oxidisers.

We apologize that the confusion between excluding archaea and later reference to AOA in line L146. Although we did not heavily analyze archaea for this study as it was very low in abundance, line 146 was included for any readers that may be wondering specifically about AOA, as they are an important component of ammonia oxidation, as you point out. Therefore, we would like to include this line to clarify that the exclusion of archaea is appropriate, not only due to their low abundance but also due to no detection of AOA. We conducted the PCR assay of AOA-amoA genes in response to this comment, but the expected amplicon products were not obtained. This confirms AOA abundance was too low to detect as found in 16S sequences.

5. L91-100 & Figure 2. Variation between communities in treatments needs testing statistically. Use of ANOSIM or similar to test for treatment and/or time-specific variation is needed. The PCoA does show clear separation of communities in the PVC treatment from other treatments, but the subsequent Figure S17 suggests a potential reason for this, namely the presence of biofilm on the surface of these sediment microcosms. Unfortunately, the experimental/sampling design has not allowed partitioning of community composition between sediment vs. microplastic (in sediment), and in the case of the PVC treatment, there would seem to be a third community present in these microcosm (the surface biofilm). Unfortunately, without physical separation of the microplastics at the end of the experiment from the sediment, it becomes impossible to determine whether differences in community composition between substrates are driven by differences in the communities in the sediment and/or on the biofilms present on each microplastic type. In the PVC treatment, there may be a third community (ie. the biofilm), and

this may drive the separation of the PVC (overall) community to diverge from those in other treatments.

Thank you for your comment on this statistical addition. We have added this and appreciate the improvement to the manuscript!

In regard to the comparison between plastic biofilm, sediment microbial community and the surface biofilm of PVC, we hope to clarify that our experimental design was *not* intended to separate these communities. Unlike Pinnell and Turner 2019, we examined the effect of small microplastics (less than 100 μm) on the total sediment microbiome and associated biogeochemical cycles. The hypothesis is that the presence of different microplastics would affect the structure and function of intact sediment communities (i.e. is sediment with plastic fundamentally different from sediment without?). Sequential changes in biofilm formation present on microplastics could then alter their surrounding community. This would result in alteration of nutrient cycling of the overall sediment community. At this juncture we realize that we cannot distinguish between organisms growing as part of the biofilm on the plastics and what is in the surrounding sediments; that is not the focus of this manuscript. Our intention is to compare the entire microbial community: inclusive of what has formed on the plastic, in surrounding sediment and, in the case of PVC, the response surface biofilm. Currently available techniques for extraction of microplastics in this size range ($\sim 100 \mu\text{m}$) from sediments are physically disruptive and would compromise the microplastic biofilm. Therefore, evaluating the biofilm separate from the sediment for plastics of this size range is not currently feasible. This would be equally challenging in a lab or field study. Furthermore, microplastics of this size range are more abundant than larger microplastics. Thus, it is critical to understand the consequences of their presence. We believe our approach, therefore, to be a synergy between environmental relevancy (size of microplastics and effects on sediment systems, using dosages that are compatible with field-derived samples) and working within experimental constraints (i.e. limiting us from separating small microplastics from sediments while preserving their biofilm).

6. L107-124 and Figure 3 and associated Supplementary Figures. What is not clear from the presentation of this data is whether any of these taxa are major components of the communities. One of the problems is that as data is averaged across replicates, we can't see whether an

individual replicate may be significantly skewing increases/decreases in a taxon within a treatment. Additionally, the increases presented could be relatively small (eg. an increase in taxon abundance from 1% to 2%), or much more important eg. a 10-fold increase of 1% to 10%. For this description of data, the order of magnitude and in particular if this relates in a taxon becoming a major component within a community is of interest and this description should drill down on those taxa which become important / abundant (or conversely) that may decline following treatment (ie. be negatively impacted by a particular polymer).

To address this, we have added relative abundance plots by sample only for family (Fig. S5), class (Fig. 4B) and phylum (Fig. 3B). The addition of Fig. S5 is referenced on lines 140-141. Although there are some differences between treatments, we feel providing more detail on these differences is unnecessary in light of the significant differences between taxa provided by our DeSeq results (Figure 3B, Figs. S6-12, Fig. 13B). This is a more robust analysis of differences between treatments than fold changes, and accounts for the variability between replicates.

7. Figure 4. Need to recognise that formation of nitrite could be via both ammonia oxidation and/or partial denitrification.

Thank you for the comment. We agree, and feel this likely explains the higher nitrite than nitrate concentrations observed, but did not adequately state this in the manuscript. This is best addressed in the discussion, where we have added “Yet, partial denitrification contributing to the NO_2^- pool could have occurred, in addition to nitrification, as evidenced by the considerably higher dissolved NO_2^- than NO_3^- .” at lines 486-488. However, as nitrate is also high, this observation does not significantly change the interpretation of our results.

8. L161 and associated text on Q-PCR. Need to determine whether differences in relative abundance of these genes is significant between treatments and/or timepoints.

The statistical information for this is provided in Table S7. Our view was that this provided sufficient detail for the reader. However, to address this comment we have added the p-values for the treatments discussed in the referenced lines, in comparison to the control (please see text on lines 229-230).

9. L166 and Figure 6 and SIFigure16 (Rate measurements). Insufficient detail is given in the

Methods to allow interpretation of the denitrification rate measurements. The authors also mention data for measurement of anammox (although there are no corresponding methods). Far more information is needed on this and additionally for anammox ^{15}N -labelling of both ammonium and nitrite is needed.

Further information has been added in brief to the methods, lines 933-940. “Briefly, 2 g of homogenized sediment in exetainer tubes was helium-purged and dark-incubated overnight to remove residual NO_2^- and NO_3^- . Six replicates of exetainer tubes per sample were amended with 100 nmoles $^{15}\text{NO}_3^-$ and then incubated at room temperature in dark. Both anammox and denitrification activities were stopped by adding saturated zinc chloride (ZnCl) solution after 0, 1 and 2 hr of incubations. Time series production of $^{29}\text{N}_2$ and $^{30}\text{N}_2$ was measured on an isotope ratio mass spectrometer and used to calculate the rate of denitrification and Anammox following Song and Tobias (2011)⁵⁵.”

10. L184 I’ve highlighted this already in Point 5 above, but the sampling regime doesn’t allow discrimination between the communities in the sediment vs. the communities that may form biofilms upon the substrates. Potentially, community shifts could be due to substrate-specific communities on the microplastics, rather than changes in the bulk sediment. As noted previously for the PVC treatment, the community shift may be driven by the sub-community present in the biofilm.

The authors appreciate this point and that this distinction is important. Nonetheless, in terms of altering bulk sediment community composition and geochemical cycling in sediments (and their subsequent impact on surrounding ecosystems) we feel this distinction is secondary. However, in light of your comments we have chosen to also address this in text, for the reader. In lines 295-299: “These sediment communities encompass both the sediment in proximity to the microplastics, as well as the biofilms thereon. This unit is relevant as it reflects the overall impact of microplastic-contaminated sediments on the aquatic ecosystem. Further, attempts to physically separate the biological constituents of the microplastics and the sediment would disrupt these communities.” And in lines 798-792: “Although the difference between biofilm communities and surrounding sediment communities cannot be determined using our approach, the outcomes between our treatments robustly illustrate the influence microplastics may have on intact sediment ecosystems.”

11. L224-229. This is problematic. Depending upon the sediment type/composition (which also needs characterisation and presenting), it is possible that denitrification may only be occurring in the top 1-5 mm. By homogenization of samples during processing, the authors have lost the ability to distinguish variation in community composition (and function), between different redox zones. (see also L237-239).

Indeed, further analysis of additional and shallower depths would have provided more detail on the redox zone; however, this was not within the objectives of this study. The authors respectfully disagree that the existing design is problematic and reference a number of published studies where the homogenized surface 1 or 2 cm are characterized:

Lisa, J. A.; Song, B.; Tobias, C. R.; Hines, D. E. Genetic and biogeochemical investigation of sedimentary nitrogen cycling communities responding to tidal and seasonal dynamics in Cape Fear River Estuary. *Estuar. Coast. Shelf Sci.* **2015**, *167*, A313–A323.

Mason, O. U.; Scott, N. M.; Gonzalez, A.; Robbins-Pianka, A.; Bælum, J.; Kimbrel, J.; Bouskill, N. J.; Prestat, E.; Borglin, S.; Joyner, D. C.; et al. Metagenomics reveals sediment microbial community response to Deepwater Horizon oil spill. *ISME J.* **2014**, No. October 2013, 1–12.

Stauffert, M.; Cravo-Laureau, C.; Jézéquel, R.; Barantal, S.; Cuny, P.; Gilbert, F.; Cagnon, C.; Militon, C.; Amouroux, D.; Mahdaoui, F.; et al. Impact of Oil on Bacterial Community Structure in Bioturbated Sediments. *PLoS One* **2013**, *8* (6).

We characterized commonly described sediment type/composition, i.e. the sediment organic N and C content, as well as sampling location, salinity and habitat type. Additional text has been added to methods, lines 867-868.

12. L229-231. This is highly speculative.

This has been modified to read: “Theoretically, *in situ* degradation of PUF may have contributed to labile inorganic N for nitrification and coupled denitrification, and should be addressed in further studies.”

13. L243. Claim of increases in sulfate reduction being a product of plastic composition needs justification.

This discussion has been expanded to compare the results provided by Pinnell and Turner (2019). We have added this discussion and introduction, and changed the sentence referenced by the reviewer to read: “As such, the increase in sulfate-reducing bacteria is yet unexplained. However, it is may be a product of the plastic composition (e.g. a shared additive between tested PVC and Pinnell and Turner (2019) PHA) or a physical response of the sediment environment (e.g. increased hydrophobicity).”

14. L247-248. If, as the authors suggest, presence of microplastics increases sediment porosity, this should lead to increased oxygenation which should stimulate (aerobic) nitrification and hence potential reduction of ammonium concentrations rather than increases.

Thank you for pointing this out, and we agree! However, we would like to clarify that the sentence the reviewer is referencing here refers to the conclusions of another paper, not our own hypothesis. We have added the following underlined text, to address the reviewer’s concern: “These authors proposed that an increase in sediment porosity allowed for greater diffusion from the sediments; yet, increasing porosity would also increase oxygen diffusion and thus nitrification, decreasing NH₄⁺.”

15. L254-255. Why should antimicrobial properties of microbeads be active against nitrifiers specifically and not against other guilds and/or taxa?

This sentence is also in reference to the hypothesis of another, similar paper (Cluzard et al. 2015) and was included for the purposes of discussion. To clarify, we have added selected citations referencing the antibiotic response of nitrifiers and sulfate-reducers. “Another hypothesis is that the microbeads possessed antimicrobial properties, which may select for certain taxa (e.g. sulfate-reducers and gram-negative³⁴) and against others (e.g. nitrifiers³⁵)³⁶.” We feel a discussion of the actual mechanism for antibiotic taxa selection is outside the scope of this paper.

16. L290-314. The long discussion on hydrocarbonoclastic taxa and potential hydrocarbon degradation is tangential to this current study and should be removed.

We recognize that this is not the focus of the study and appreciate this suggestion from the reviewer. As hydrocarbon degradation is still tangential to the study (denitrification, selection of a biopolymer) and similarly addressed in other plastic biofilm papers (e.g. Pinnell and Turner

2019), we believe some discussion is necessary for readers interested in this topic. Heeding your advice, however, we have significantly reduced this section.

17. L343. I'm unclear as to whether there was any attempt to sterilise plastics going into the experiments. For example, for the PUF, from the description, I'm assuming this is used-polymer and potentially already colonised by microorganisms.

We appreciate this question, and the potential implications it could have to our study. The plastics were not sterilized. However, this was not done for several reasons. First, any sterilization techniques (autoclave, chemical, etc.) may have altered the chemical nature and physical form of the plastics we used, as they are such small microplastics and easily affected by harsh techniques. In addition, we are confident that contamination was not a problem as attempts to extract DNA from plastics in our lab has been extremely unsuccessful in other studies, supporting no initial microbial contamination. Also, in the context of environmental relevance, microplastics that reach aquatic ecosystems have not been autoclaved.

18. L355. Needs characterisation and description of sediment to better understand experimental system.

This was addressed in comment 11. If the reviewer would like further detail, please specify.

19. L401-402. Normalization to Q-PCR 16S rRNA gene numbers is not clear.

We agree. We have changed this to be a simple ratio to 16S rRNA. This has been updated in the figures and source data file.

20. References. Some references are incomplete eg lack page numbers. Others have inconsistent format usage (eg. inconsistent use of capitals in article titles), use of DOI rather than volume/page numbers for long-published papers (eg. Ward, 2008), gene names need italicizing etc.). Please review/revise all references.

This has been modified accordingly.

Reviewer #2 (Remarks to the Author):

Author provides an interesting experiment about adhered microorganisms on MPs and nitrifying and denitrifying activities under exposure experiment. Similar exposure experiment has been addressed in previous reference (Wei et al., 2019). Therefore, I don't think it is novel and sound enough to publish in Nature Communication

Thank you for your time in reviewing our paper. Wei et al. (2019) concerns anaerobic digestion in waste activated sludge, an entirely different model system than our study. As such, it was not germane enough to even include in our references and does not compromise the novelty of our study. Indeed, the novelty of our study is one of the most exciting aspects, highlighting potential impacts on marine communities and future research needs in the field.

Below, I only outline some major concern.

Line 341-353. Why author used commercial plastics in present study and was it consistent with environmental plastic debris in terms of weathering, composition and size range?

I think it would be more convincing.

We appreciate the reviewer's view on this point. Where possible, the authors chose to use commercially available plastic products (from which we generated microplastics of a select size range) as such materials are the most environmentally relevant precursors of secondary microplastics to the environment. This is in contrast to other studies wherein the authors purchased plastic 'beads', made to be perfectly spherical and homogeneous in size. In some cases, authors have used non-representative polymers such as ultra high molecular weight polyethylene or even unknown polymer types. Such materials are less relevant to the bulk of microplastics present in the environment. Microplastics in the environment consist of a virtually infinite range of weathered condition, composition and size. Thus, inclusion of all in a study is impossible. We therefore chose to include a range of polymer types and a reasonable choice of fragment size.

Line 354 incubation was performed in the lab with too many controlled factors. It is hard to believe to reveal the actual reaction.

In the marine environment an infinite range of conditions exist. To further understanding, scientists must seek to identify cause and effect and thus must control conditions. As this is the first study to address the total sediment microbial and nitrogen cycling responses to microplastic pollution, a controlled experiment was a required starting point. Indeed, we feel it provided novel data from which to design futures studies intended to explore additional variables and to provide insights on environmentally contaminated locales.

Minor comment

Line 27-30. This sentence should be reorganized to clearly present the intended meaning

This comment has been addressed in the text. The sentence now reads: “Although many plastics are inherently buoyant, microplastics can be exported to sediments after biofouling or incorporation into marine snow or fecal pellets^{5,6}.”

Reviewer #3 (Remarks to the Author):

The manuscript by Seeley et al analyzed the bacterial community structure as well as abundance and activity of N-cycling bacteria in a salt marsh sediment in response to additions of different plastic materials. The authors performed a microcosm experiment adding four different types of plastic to the incubations and determined the rates for potential denitrification and the anammox process.

Molecular analyses (16S sequencing, qPCR for bacterial amoA, nirS, nirK) have been used to investigate the relative abundance of prokaryotic taxa and to quantify corresponding microbial guilds involved in N-cycling.

The study is a valuable contribution to our understanding of the relative impact of different (micro) plastics on bacterial community structures and overall N-cycling in salt marsh ecosystems.

We would first, like to thank you for taking your time to review our paper. We found your insights to benefit the paper greatly, and appreciate you bringing these edits to our attention. We have addressed each of your points individually, below.

Comments:

- The experimental set up is not complicated, therefore I think Fig. 1 is not essentially needed. Move to supplement.

We appreciate the comment from the reviewer. However, we prefer to keep the figure in text as it is useful in the visualization of study.

- In Table 1 it would be good to include the errors for the indices. That way its easier to understand differences between days and treatments.

The standard error has been added to the table.

- line 111: In the figure caption please correct to: “Families were determined to be significantly different between treatments...”

This has been changed to “Families that are significantly different in relative abundance between treatments and controls (averaged across collection dates), determined using DeSeq ($\alpha = 0.01$).”

- Figures 4, 5, S13, S14, S15, S16: If I am not mistaken the minimal number of data points for a boxplot is 9. If you have less, you should rather show all individual data points as the visualisation as a mean with spreads is misleading. Please also make sure to always include a statement on the number of n in the figure legends. This is missing in a few cases. In case of S16, where n=1, I recommend a table instead.

We thank the reviewer for pointing this out. We have changed these figures to bar graphs. For Figure S16, however, we have left it as a bar graph to facilitate comparison to the N & C content in sediments.

- As an addition to Fig. 4 I suggest to calculate the N-balance with the data on inorganic N. This way you could see if any substantial amounts of N were lost from your system (e.g. through denitrification) or gained (e.g. through N mineralization). Also, have the initial values been measured? If so, please include in this graph it may make trends/ dynamics more clear.

Thank you for the suggestion. The experimental design does not allow calculation of the N-balance between different treatments. Whole core experiments, measuring the fluxes of DIN and N_2 , are typically used to measure denitrification efficiency and N mineralization. However, we used the modified method of calculating denitrification efficiency recently published by Semendo and Song (2020) and estimated nitrogen removal efficiency based on the rates of anammox and denitrification and DIN concentrations among the treatments (Fig. S17). The PUF treatment had the highest denitrification efficiency with the greatest rates of denitrification potentials. The 2nd highest efficiency was found in the PLA treatment. PVC DIN removal was comparable to the control, much lower than the other plastic treatments. We have also added information on this in the text (lines 274-277): “Further, the potential rates of denitrification and

anammox were compared to total DIN, following Semedo and Song (2020) to estimate DIN removal capacity. This revealed that PVC and the control treatments had the lowest DIN removal capacity, while PLA and PUF had the highest (Fig. S17).²⁴ Further, initial values were calculated for all parameters except for rate estimates. However, because they are an individual point (n = 1) we cannot calculate error. Therefore, we chose not to include these on the graphical representations. To improve clarity, however, we have added the initial concentrations to the legend of each figure.

- Fig 5: I would prefer to see the data for the initial time point in the graph rather than wrapping my head around a double ratio, which is what the normalization to 16S and initial community is I presume. So please just show all 16S normalized data.

We agree. We have changed this to be a simple ratio to 16S rDNA. This has been updated in the figures.

- Fig. 6: Please include the unit for the y-axis directly at the axis. Also, add letters to indicate sig. differences to the control.

The axis title has been modified and there are now asterisks to depict significant differences with the control.

- line 223-224: How do you arrive at this statement? The conditions for the potential denitrification assay were quite different than the conditions of the microcosm experiment.

To address this, we have included the underlined text: “Not only were NO_3^- and NO_2^- concentrations in the overlying water extremely low, but the PVC treatment sediment also exhibited the lowest relative abundance of *nirS* gene and lowest potential rate of denitrification in the post-incubation sediment slurry experiment.”

- line 233: Please correct: “... in the overlying water extremely low, ...”

Thank you. We have added “water” to the sentence.

- line 238: Please rephrase to something like: Most genera of [...] showed a significantly higher relative abundance in... This should be carefully corrected in other places of the manuscript too.

Thank you for pointing this out. We have corrected it here and elsewhere in the manuscript.

- In the method section the description of the qPCR assay for the amoA gene is missing. Please amend.

Thank you for bringing this to our attention! The methods have been adjusted.

- line 439-441: In the listed reference NO₂⁻ and NO₃⁻ are not measured separately, but in this manuscript there is data specifically for both. Furthermore, PO₄³⁻ is not mentioned in the reference at all. Please correct and mention method for determination of NO₂⁻, NO₃⁻ and PO₄³⁻ quantification.

The reference has been changed to encompass all these nutrients.

- Fig. S1: Please add error bars and indicate from which day of the incubation the data was obtained.

We have updated the figure and caption, accordingly.

REVIEWERS' COMMENTS:

Reviewer #1 (Remarks to the Author):

The authors have provided appropriate responses to the comments of the reviewers, and I consider the revised manuscript is now suitable for consideration for acceptance by the editor. Thank you to the authors for carefully considering and responding to the comments of all reviewers.

Reviewer #2 (Remarks to the Author):

Indeed, lots of previous studies used plastic microbeads during exposure experiment, which is obviously contrast to the realistic environment. In fact, fibrous MPs, especially for PET type, were most abundance in the ocean. I appreciate the efforts for artificial MPs manufacture, but still, my major concern is that authors still used completely different artificial MPs with MPs in the field although authors keep stressing the similarity of these samples. Below, I would like to summarize polymer composition in field and hope authors may consider using these MPs (Table.1). Besides, environmental MPs would be constantly influenced by surroundings and their properties, especially surface of MPs were totally different from those new manufactured MPs because of the physical-chemical interaction with environment and attached biofilm. From this point, I do not think these artificial MPs could be treated as environmental MPs. At least, authors should expose this MPs in the environment for several days prior to exposure experiment and I found several studies using this method, especially for exposure experiment. In that way, this research would convince most of potential readers that environmental MPs would influence the nitrogen cycling as authors claimed. I think this paper should be published somewhere else, not in Nature communication. I am sorry, but I have to reject this paper at this time.

Reviewer #3 (Remarks to the Author):

I have no further comments and am content with the revised version of the manuscript.

Comments to reviewers for “Microplastics affect sedimentary microbial communities and nitrogen cycling” by Meredith Seeley, Bongkeun Song, Renia Paise and Robert C. Hale

Reviewer #1 (Remarks to the Author):

The authors have provided appropriate responses to the comments of the reviewers, and I consider the revised manuscript is now suitable for consideration for acceptance by the editor. Thank you to the authors for carefully considering and responding to the comments of all reviewers.

Thank you for your previous suggestions and approving of the changes we made in response. We appreciate your time and efforts, and your contribution to improving our manuscript!

Reviewer #2 (Remarks to the Author):

Indeed, lots of previous studies used plastic microbeads during exposure experiment, which is obviously contract to the realistic environment. In fact, fibrous MPs, especially for PET type, were most abundance in the ocean. I appreciate the efforts for artificial MPs manufacture, but still, my major concern is that authors still used completely different artificial MPs with MPs in the field although authors keep stressing the similarity of these samples. Below, I would like to summarize polymer composition in field and hope authors may consider using these MPs (Table.1).

Besides, environmental MPs would be constantly influenced by surroundings and their properties, especially surface of MPs were totally different from those new manufactured MPs because of the physical-chemical interaction with environment and attached biofilm. From this point, I do not think these artificial MPs could be treated as environmental MPs. At least, authors should expose this MPs in the environment for several days prior to exposure experiment and I found several studies using this method, especially for exposure experiment. In that way, this research would convince most of potential readers that environmental MPs would influence the nitrogen cycling as authors claimed.

I think this paper should be published somewhere else, not in Nature communication. I am sorry, but I have to reject this paper at this time.

Thank you for taking the time to re-review our paper, and provide these sources. Indeed, we recognize that microplastics exist in an immense diversity of polymer types, morphologies and sizes in the environment. The characteristics of microplastics reported in the published literature are evolving as methods improve and sampling of additional environmental compartments increase (for example, progressing from water surface to sediments). The composition of microplastics in sediments is expected to differ from those in the water column, in part due to differences in plastic density and settling behavior. Interestingly, three articles cited in Table 1 (Lahens et al; Mai et al; Xu et al) focus on microplastics in the water column, not in sediments.

Also, with the exception of PLA, other three polymer types (PE, PVC, PUF/PUR) actually appear in your list of polymer types. PLA is important to examine as it is a biopolymer, a possible “environmentally green” replacement. Hence, data on its fate is both critical and lacking to date.

Our study intentionally chose four diverse/representative polymer types, a common particle size range and an environmentally realistic concentration range. We hypothesized that polymer composition would drive differences in microbial community response. Thus, examining a series of four selected polymer compositions (and control) was a deliberate element of our experimental design. We feel this is justified, especially as this is a foundational study. In other words, the polymer types were selected intentionally to encompass a range of polymer compositions. Abundance of polymer types from site to site in the environment. We feel attempting to mimic a hypothetical “composite” composition, based on our presently incomplete and inaccurate data, would not be beneficial.

To address this concern for you and our readers, we’ve added the following text to the manuscript. We hope you find it satisfactory in addressing your concerns, and thank you very much for your time in reviewing our manuscript!

Lines 278-279: “In summary, all four treatment plastics evaluated here differ in physical and chemical characteristics, hence their selection for this study.”

Lines 440-441: “This is foundational for future work assessing risk of microplastic pollution in different environments.”

Lines 488-491: “Further, being a foundational study in the field, the plastics we selected were not intended to be representative of all environmental sediment microplastic pollution, but rather to embody a range of the characteristics that may be encountered.”

Reviewer #3 (Remarks to the Author):

I have no further comments and am content with the revised version of the manuscript. Thank you for reviewing our manuscript, and making suggestions that have improved our paper. We appreciate your contribution and time!